# *Weissella cibaria* suppresses colitis-associated colorectal cancer by modulating the gut microbiota-bile acid-FXR axis

Qiuyao Hao,[1,2] Fei Huang,[1,2] Liangzheng Chang,[2] Hongyuan Dai,[2] Weiwei Chen,[3] Yiran Yao,[4] Yunhuan Zhen[1,2]

**ABSTRACT** Gut microbiota dysbiosis critically contributes to colitis-associated colorectal cancer (CAC) pathogenesis, positioning microbial modulation as a promising therapeutic strategy. *Weissella cibaria* (*W. cibaria*) is an emerging probiotic with potential cancer-inhibiting effects. This study investigates the anti-tumorigenic potential of *W. cibaria* in an azoxymethane/dextran sulfate sodium-induced CAC murine model. Mice were orally administered *W. cibaria* every 2 days from the beginning of the model construction until the end of the experiment. The study demonstrated significant changes in the gut microbiota of CAC mice, with a significant increase in the relative abundance of *Lactobacillaceae*. Supplementation with *W. cibaria* restored the intestinal barrier and significantly reduced the relative abundance of *Lactobacillaceae* in the gut microbiota. The changes in the gut microbiota reduced bile salt hydrolase activity and unconjugated bile acid (BA), reversing tumorigenesis in CAC mice. Changes in intestinal BA after *W. cibaria* supplementation upregulated farnesoid X receptor (FXR) expression in the intestine of CAC mice and inhibited the nuclear factor kappa-B pathway. Our findings establish that *W. cibaria* mitigates CAC progression through the gut microbiota-BA-FXR axis, providing mechanistic evidence for its probiotic application in CAC prevention and therapy.

**IMPORTANCE** Chronic gut inflammation driven by microbiota dysbiosis is a pivotal contributor to colitis-associated colorectal cancer (CAC) pathogenesis. Emerging evidence highlights *Weissella cibaria* (*W. cibaria)* as a promising anti-colorectal cancer agent (Y. Du, L. Liu, W. Yan, Y. Li, et al., Sci Rep 13:21117, 2023, https://doi.org/10.1038/s41598-023-47943-7; S. Ahmed, S. Singh, V. Singh, K. D. Roberts, et al., Microorganisms 10:2427, 2022, https://doi.org/10.3390/microorganisms10122427), yet its role in CAC remains unexplored. To address this gap, we investigated the inhibitory effects of *W. cibaria* on CAC development *in vivo* and elucidated its underlying mechanisms. Our results demonstrated that oral administration of *W. cibaria* significantly reshaped gut microbial communities and activated bile acid (BA)-related metabolic pathways. Subsequent mechanistic studies revealed that microbiota remodeling by *W. cibaria* altered intestinal BA composition, particularly activating the farnesoid X receptor (FXR). FXR activation mediated by these BA shifts was identified as a critical suppressor of tumorigenesis, establishing *W. cibaria* as a novel probiotic capable of attenuating CAC progression. Collectively, this study uncovers a protective axis linking *W. cibaria*-driven microbiota modulation, BA metabolism change, and FXR-dependent tumor suppression, providing experimental evidence for probiotic-based CAC intervention strategies.

**KEYWORDS** *Weissella cibaria*, bile acid, FXR, colitis-associated colorectal cancer, gut microbiota

**Peer Reviewer** Ipsita Mohanty, University of California San Diego Skaggs School of Pharmacy and Pharmaceutical Sciences, La Jolla, California, USA

Address correspondence to Yunhuan Zhen, yunhuanzhen72@163.com, Yiran Yao, yaoyiran724@163.com, or Weiwei Chen, weiweichen@gmc.edu.cn.

Qiuyao Hao and Fei Huang contributed equally to this article. Author order was determined by drawing straws.

The authors declare no conflict of interest.

Colorectal cancer (CRC), the third most prevalent malignancy globally, is increasingly characterized by earlier onset and rising mortality, particularly in colitis-associated

CRC (CAC) linked to inflammatory bowel disease (1, 2). Mounting evidence implicates gut microbiota dysbiosis as a key driver of CRC pathogenesis. Fecal microbiota transplantation studies demonstrate that CRC patient-derived microbiota promotes tumorigenesis in both germ-free and conventional mice, underscoring the microbiota's causal role in intestinal carcinogenesis (3). Further work establishes that dysbiotic microbiota fuels chronic inflammation, accelerates CAC progression, and facilitates metastasis (4, 5). It can be seen that regulating gut microbiota dysbiosis becomes a therapeutic measure for CAC.

Probiotic interventions, particularly *Lactobacillus* and *Bifidobacterium* species, show promise in restoring microbial balance and suppressing CRC in preclinical models (6, 7). Emerging research highlights bile acids (BAs) as critical mediators linking gut microbiota to intestinal pathophysiology. Gut microbes enzymatically reshape the BA profile via bile salt hydrolase (BSH), 3-, 7-, and 12-hydroxysteroid dehydrogenase, and other modifications (8, 9). Research suggests that imbalanced gut microbiota alters the BA profile and can promote the development and progression of CRC (10). This microbial-BA cross talk suggests that probiotics mitigating dysbiosis could therapeutically recalibrate BA metabolism to inhibit carcinogenesis.

*Weissella cibaria* (*W. cibaria*) belongs to the genus *Lactobacillus* and can be extracted from Korean kimchi, some fermented foods, and human feces (11). Compared to conventional probiotics like *Lactobacillus rhamnosus* GG, *W. cibaria* demonstrates superior nuclear factor kappa-B (NF-κB) pathway suppression and pro-inflammatory cytokine reduction (12, 13). Moreover, *W. cibaria* inhibits colorectal cancer cell growth by inducing apoptosis (14). In addition, *W. cibaria* was more abundant in normal human feces than in CRC patients (15). However, its capacity to modulate the microbiota-BA-farnesoid X receptor (FXR) axis—a key regulatory circuit connecting microbial metabolism to intestinal inflammation and tumorigenesis—remains unexplored in CAC.

This study demonstrates that *W. cibaria* supplementation attenuates tumor burden in azoxymethane/dextran sulfate sodium (AOM/DSS)-induced CAC mice by orchestrating a triad of effects: restructuring gut microbiota to reduce *Lactobacillaceae* enrichment, reducing BSH activity and unconjugated BA accumulation, and activating intestinal FXR signaling to suppress NF-κB-driven inflammation. By mechanistically linking microbial shifts to BA-mediated FXR activation, our work establishes *W. cibaria* as a novel therapeutic candidate targeting the gut microbiota-BA-FXR axis in CAC prevention.

## MATERIALS AND METHODS

### Mice

Six-week-old C57BL/6J and homozygous intestinal-specific conditional FXR knockout (FXR$^{cKO}$) male mice weighing 20 ± 2 g were used in the study. C57BL/6 mice were purchased from Special Pathogens Free (SPF) Biotech (Beijing, China). Mice were housed in the SPF Animal Laboratory of the Fifth Medical Center of the General Hospital of the People's Liberation Army at 20°C–26°C, 50% relative humidity, and a 12 light-dark cycle. Food and water were provided at random.

### CAC mouse construction and intervention methods

Construction of a mouse model of colorectal cancer using the AOM/DSS approach. In this study's experimental protocol, AOM (10 mg/kg, Sigma-Aldrich) dissolved in saline was injected intraperitoneally into all groups of mice except the Control group. After 7 days, DSS (2%, wt/vol, MP Biomedicals) was added to the mice's sterile drinking water for 1 week and then changed to normal sterile drinking water for 2 weeks. There was a DSS cycle every 3 weeks, and three cycles were repeated. From the first day of model induction, mice were given 200 µL (PBS or $1 \times 10^9$ CFU *W. cibaria*) by gavage every 2 days until the end of the experiment. All mice were euthanized after day 84. According to previous studies, colon tumors were classified into three groups based on size: small

tumors (<1 mm), medium tumors (1 mm ≤ size ≤ 2 mm), and large tumors (>2 mm). Tumor load was calculated using the following formula: tumor load = (number of small tumors × 1) + (number of medium tumors × 2) + (number of large tumors × 3) (16). Tumor load represents a dimensionless composite metric derived from tumor number and size stratification, where elevated scores correlate with increased neoplastic burden.

## Breeding knockout mice

H11-Vil1-iCre (a codon-optimized Cre recombinase variant) mice and Fxr$^{fl/fl}$ (conditional knockout mice with loxP sites inserted on both sides of the FXR gene) mice were generated by GemPharmatech Co., Ltd. (China). Primers used to identify mice are shown in Table S1. All mice were maintained in the SPF Animal Laboratory of the Fifth Medical Center of the General Hospital of the People's Liberation Army. Eight-week-old H11-Vil1-iCre mice were crossed with 8-week-old FXR$^{fl/fl}$mice to obtain H11-Vil1-iCre/FXR$^{fl/fl}$mice as FXR$^{cKO}$mice. Then, we demonstrated the breeding of FXR$^{cKO}$ mice by examining the protein expression of FXR in their intestines (Fig. S1). Although H11-Vil1-iCre mediates FXR deletion in both small intestinal and colonic epithelia, all subsequent analyses were performed exclusively on colon tissues.

## Histopathological analysis and immunohistochemistry

Mouse intestinal tissues were fixed in formalin, paraffin-embedded, and then cut into 4 µm thin slices placed on slides. The tissues were stained with hematoxylin-eosin (HE) and subsequently observed under the microscope for tissue morphology.

For tissue immunohistochemical staining, treated tissue specimens were placed at 95℃ for 3 min using sodium citrate buffer for antigen repair. Subsequently, sections were incubated with anti-Ki67 (ab15580, Abcam), anti-ZO-1 (GB111402, Wuhan, ServiceBio), and anti-Occludin (GB111401, Wuhan, ServiceBio) primary antibodies at 4℃ overnight. Subsequently, the secondary antibody coupled with horseradish peroxidase was incubated at room temperature for 30 min. The latter was reacted with diaminobenzidine, and the color of the antibody was observed under a microscope. The number of positively stained cells was counted using ImageJ software.

## Detecting intestinal permeability

Mice were deprived of water and food for 12 hours. The next day, mice were gavaged with 40 mg of FITC-Dextran (FD4, Sigma-Aldrich) per 100 g of body weight. Blood was collected from the medial canthus vein of mice after 4 hours, and they were protected from light throughout the operation. Serum was separated by centrifugation at 1,500 $g$ for 10 min at 4℃ and diluted 1:2 with PBS. Black flat-bottom 96-well plates were selected, and 100 µL of diluted serum was added to each well (three replicate wells per group). Then, a standard curve was generated. Optical density (OD) was measured at 490 nm excitation light and 520 nm emission light.

## Culturing of bacteria

*W. cibaria* (BJMCC14295) was bought from Beijing Microbiological Culture Collection Center (Beijing, China) and cultured in MRS (M8330; Solarbio, Beijing, China) broth media at 37℃ for 24 hours. Afterward, 200 µL of bacterial solution (1 × 10$^9$ CFU) was prepared for gavage administration to mice.

## Enzyme-linked immunosorbent assay

Mouse colon tissue protein extracts were prepared. The activity of interleukin-1 beta (IL-1β), interleukin-6 (IL-6), and tumor necrosis factor alpha (TNF-α) in colon tissue was detected following the protocol provided in the enzyme-linked immunosorbent assay (ELISA) kit (Dakewe, Shenzhen, China).

## Quantitative analysis of *W. cibaria*

A real-time quantitative polymerase chain reaction (RT-qPCR) technique was used to determine the colonization level of *W. cibaria* in the mouse colon. To establish a standard curve, serial dilutions of leech suspensions were made in sterile PBS to a final concentration of $10^2$–$10^8$ CFU/mL, and genomic DNA was extracted from each dilution using a bacterial genomic DNA extraction kit (DP302, Tiangen, China, Beijing). For experimental samples, a QIAamp Fast DNA Stool Mini Kit (Qiagen, Hilden, Germany) was used to isolate DNA from the colon contents of each group of mice. qPCR was performed on a QuantStudio 6 Real-Time PCR System (Thermo Fisher Scientific) using the Taq Pro U + Multiple Probe qPCR Mix (QN213-01, Vazyme, Nanjing, China). The primers and double-labeled probe sequences used are shown in Table S2. A standard curve was made by plotting threshold cycle (CT) values against $\log_{10}$-transformed CFU/mL counts (*W. cibaria*: $10^2$–$10^8$ CFU/mL). The CT values of the colon contents of each group were used in this equation for CFU/mL transformation to obtain the level of *W. cibaria* colonization in each group (17, 18).

## 16S rRNA gene sequencing

After collecting the colonic contents of mice, the DNA in the colonic contents was extracted using HiP Stool DNA Kits (Magen, China), and the quality of the DNA was assessed. The V3-V4 region of the 16S rRNA gene was then amplified by PCR using specific primers. The library quality was checked using the ABI StepOnePlus Real-Time PCR System with NovaSeq 6000 onboard sequencing. Significant *P*-values associated with linear discriminant analysis with effect size (LEfSe) were corrected for multiple hypothesis testing using the Benjamini and Hochberg false discovery rate correction. Data analysis and image production were performed using the Omicsmart online platform (http://www.omicsmart.com) and GraphPad Prism 10.

## Application of databases

GEPIA2 (http://gepia2.cancer-pku.cn/) is a tool for gene expression analysis based on The Cancer Genome Atlas and Genotype-Tissue Expression databases (19). We used this database to analyze and compare FXR gene expression in tumor and non-tumor tissues in patients with CRC (data from colon adenocarcinoma [COAD] and rectum adenocarcinoma [READ] were chosen for this analysis as the database splits CRC into COAD and READ).

The cBioPortal (https://cbioportal.org) utilizes a genomic database that integrates multiple types of genomes. We used data on FXR expression and survival time from 594 CRC patients to analyze the association between FXR expression and patient prognosis. Overall survival was assessed using the Kaplan-Meier method.

## Human tissue specimens

The patients who participated in this study were diagnosed with CRC by histologic examination.

## Bile acid analysis

The colonic contents of mice were mixed, weighed, and placed in 2 mL centrifuge tubes, and 500 µL acetonitrile was added. The mixture was vortexed for 1 min, ground for 5 min, allowed to rest at −40℃ for 60 min, and centrifuged at 18,000 *g* for 10 min. The supernatants were concentrated to near dryness. Then, the sample was re-dissolved in 0.2 mL of methanol (containing 50 ng/mL of internal standard), vortexed for 1 min, and centrifuged at 18,000 *g* for 10 min at 4℃; the supernatant was collected and subjected to liquid chromatography-tandem mass spectrometry analysis. Bile acid metabolites were identified and quantified by Waters Acquity UPLC (Waters Corp.) or AB SCIEX 5500 (AB Sciex) system. An Acquity UPLC BEH C18 column (100 × 2.1 mm; 1.7 µm; Waters Corp.)

was used for chromatographic separation. The mobile phase consisted of 0.05% formic acid in water (A) and 0.05% formic acid in acetonitrile (B). Column temperature was maintained at 40°C, and the flow rate was 0.3 mL/min. The gradient elution was started from 10% B for 1 min, increased linearly to 40% B over 1 min, then to 45% A over the next 3 min, followed by 60% B over the next 2.5 min, and 65% B over the next 2 min. It was further increased to 80% B over the next 2 min and held at 80% B for 2 min. Finally, the gradient was increased to 10% B over the next 0.5 min and held at 10% B for 2 min. Mass spectrometry conditions: ion source was electrospray ionization, curtain gas was 35 arb, collision gas was 7 arb, ion spray voltage was 4,200 V, temperature was 450°C, ion source gas1 was 35 arb, and ion source gas2 was 35 arb. The data were obtained and integrated using MultiQuant software, and each bile acid content was calculated using the standard curve method. Abbreviations and full names of bile acids analyzed in the text are shown in Table S3.

## RNA sequencing analysis

Fresh intestinal tumor tissues were collected from the PBS group and *W. cibaria* group mice from the CAC model. After the extraction of total RNA, eukaryotic mRNA was enriched with Oligo(dT) beads. Enriched mRNA was fragmented into short fragments and reverse transcribed into cDNA using the NEBNext Ultra RNA Library Prep Kit for Illumina (neb# 7530, New England Biolabs, Ipswich, MA, USA). The purified double-stranded cDNA fragment was end-repaired by adding the "A" base and ligated to the Illumina Sequencing Adapter. Polymerase chain reaction (PCR) amplification was performed, followed by sequencing of the resulting cDNA using Illumina Novaseq 6000. DESeq2 was used to analyze the differential gene expression between the PBS and *W. cibaria* groups. The *P* value of <0.05 and an absolute multiplicity of difference ≥1.5 were considered significant.

## RNA extraction and RT-qPCR

Total RNA was extracted from mouse colon tissue samples using FreeZol Reagent (R711-01, Vazyme, Nanjing, China) and then reverse transcribed to cDNA using PrimeScript RT Reagent Kit with gDNA Eraser (Perfect Real Time) (RR047A, Takara). TB Green Premix Ex Taq II FAST qPCR (CN830A, Takara) was used for qPCR analysis. Relative mRNA expressions of target genes were normalized to β-actin and analyzed using the $2^{-\Delta\Delta Ct}$ method. Primer sequences are listed in Table S4.

## Western blotting

Tissue proteins were extracted using RIPA buffer (HX1862, Huaxingbio, Beijing, China) with the addition of protein inhibitors and phosphatase inhibitors (GRF103, Epizyme, Shanghai, China). Tissue protein was quantified using the Bicinchoninic Acid (BCA) Protein Assay Kit (HX18651, Huaxingbio, Beijing, China). Protein samples after membrane transfer were blocked at room temperature with 5% skimmed milk for 2 hours. After sealing, the membrane was incubated with anti-FXR antibody (A24015, ABclonal, 1:1,000 dilution, Wuhan, China), anti-ZO-1 antibody (A11417, ABclonal, 1:1,000 dilution, Wuhan, China), anti-Occludin antibody (A24601, ABclonal, 1:1,000 dilution, Wuhan, China), anti-p65 antibody (8242, CST, 1:1,000 dilution, USA), anti-p-p65 antibody (3033, CST, 1:1,000 dilution, USA), anti-β-actin antibody (6009-1-Ig, Proteintech, 1:1,000 dilution, Wuhan, China) at 4°C for 12 hours. Subsequently, the membrane was incubated with horseradish peroxidase-coupled anti-rabbit or anti-mouse secondary antibodies (HX2031, HX2037, Huaxingbio, 1:10,000 dilution, Beijing, China). Then chemiluminescence was detected.

## BSH activity of colonic contents

BSH activity of colonic contents was assayed according to a previously described method (20, 21). In this experiment, the BSH enzyme activity was assayed by quantifying the

free taurine released from taurodeoxycholic acid (TDCA) after hydrolysis of TDCA by BSH, and the free taurine can bind to ninhydrin. Briefly, 50 mg of colonic contents was taken and added to RIPA buffer (HX1862, Huaxingbio, Beijing, China), ground, ice-dissolved, and centrifuged to extract total fecal protein. Then, the BCA Protein Assay Kit (HX18651, Huaxingbio, Beijing, China) was used for protein quantification. The extracted protein was then diluted to 1 mg/mL in 3 mM sodium acetate. A volume of 170 µL of 3 mM sodium acetate and 20 µL of 1 mM taurodeoxycholic acid was then added to 10 µL of the fecal protein extract and incubated at 37°C for 30 min before the samples were placed on dry ice to stop the reaction. Next, the samples were thawed and centrifuged, and 20 µL of the supernatant was mixed with 80 µL of distilled water and 1.9 mL of ninhydrin reagent (0.5 mL of 1% [wt/vol] ninhydrin dissolved in 0.5 M sodium citrate buffer, pH 5.5; 1.2 mL of 30% [wt/wt] glycerol; and 0.2 mL of 0.5 M sodium citrate buffer, pH 5.5), mixed thoroughly, and boiled for 15 min. The sample was allowed to cool, and the absorbance was determined at 570 nm. The $OD_{570\,nm}$ values were converted to BSH activity using a standard curve generated with known taurine concentrations.

## Statistical analysis

All clinical samples and animal data used in this study were analyzed. Two-group comparisons were performed using Student's $t$-tests or non-parametric tests. Multiple-group comparisons were performed using one-way ANOVA followed by Tukey's multiple comparisons. When the characteristics of the sample data meet the assumption of normal distribution, one-way ANOVA is used. If the normal distribution assumption is not satisfied, the Kruskal-Wallis test is used. All figures were presented as mean ± standard deviation. Statistical significance was defined as $P < 0.05$. *$P < 0.05$; **$P < 0.01$; ***$P < 0.001$; and ****$P < 0.0001$. ns, not significant. All statistical analyses were performed using GraphPad Prism 10 (GraphPad, San Diego, CA, USA).

## RESULTS

### W. cibaria inhibits tumorigenesis

CAC mouse models were established and administered *Weissella cibaria* or PBS via oral gavage every 2 days (Fig. 1A). Administration of *W. cibaria* significantly reduced intestinal tumor number compared to PBS-treated controls (Fig. 1B). Monitoring demonstrated that *W. cibaria* attenuated body weight loss and extended overall survival in CAC mice (Fig. 1C). Tumor burden metrics—including tumor number, size, and load—were markedly decreased in the *W. cibaria* group, and intestinal shortening was relieved (Fig. 1D through G). Histopathological analysis via H&E staining revealed amelioration of dysplastic hyperplasia in the intestine of CAC mice after *W. cibaria* treatment (Fig. 1H). Immunohistochemistry (IHC) further indicated suppressed antigen Kiel-67 (Ki-67) protein expression in both tumor-adjacent and neoplastic tissues of the *W. cibaria* group, reflecting reduced cellular proliferation (Fig. 1H). ELISA showed that the expression levels of pro-inflammatory factors IL-1β, IL-6, and TNF-α were significantly lower in the colon tissue of the *W. cibaria* group compared to the Pbs group (Fig. 1I). Collectively, these data establish that *W. cibaria* reduces inflammatory factors and suppresses tumorigenesis in the CAC model, highlighting its role in modulating both the inflammatory microenvironment and neoplastic progression.

### W. cibaria restores intestinal barrier integrity

The integrity of the intestinal barrier plays a critical role in CRC progression (22). To assess intestinal barrier dysfunction in our CAC model, we measured intestinal permeability via FITC-dextran assay. Serum FITC-dextran levels were significantly reduced in the *W. cibaria* group compared to the Pbs group (Fig. 2A), indicating that probiotic intervention effectively restored intestinal barrier integrity. To further investigate barrier restoration mechanisms, we analyzed key tight junction proteins: zonula occludens-1 (ZO-1), Occludin, and Claudin-1, established biomarkers of intestinal epithelial integrity (23).

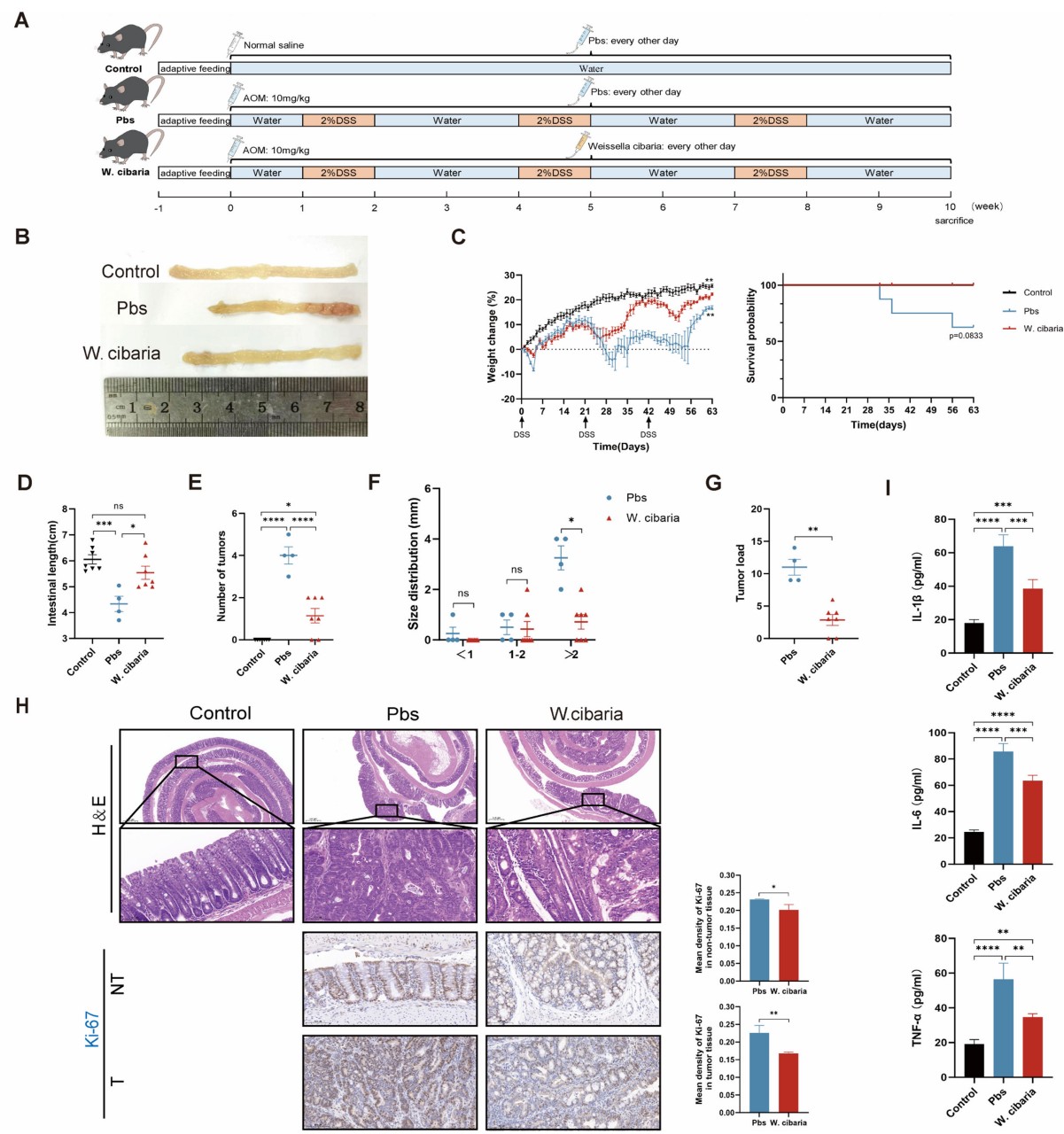

**FIG 1** *W. cibaria* inhibits the development of CAC (*n* = 4–7). (A) Animal experiment protocol (*n* = 7). (B) Macroscopic view of the colon. (C) Percentage change in body weight for each group of mice based on initial body weight (first feeding of 2% DSS water) and overall survival time of mice. (D) Intestinal length. (E through G) Tumor number, tumor size distribution, and tumor load in the colon of mice (tumor load values were calculated using the formulas provided in Materials and Methods and tumor size distribution data from panel F). (H) Representative mouse intestinal H&E staining images (scale bar = 500 and 100 µm) and Ki67 immunohistochemistry plots and bars of mice. (I) ELISA was performed to measure IL-1β, IL-6, and TNF-α expression levels in colon tissues (*n* = 4). Data are expressed as mean ± SEM. For comparisons between two groups, two-tailed unpaired Student's *t*-tests were used. For multi-group comparisons, one-way ANOVA followed by Tukey's *post hoc* test was applied. *$P < 0.05$; **$P < 0.01$; ***$P < 0.001$; ****$P < 0.0001$; ns, not significant.

Compared with Pbs mice, significantly higher levels of ZO-1, Occludin, and Claudin-1 mRNA were observed in the colon tissue of the *W. cibaria* group (Fig. 2B). Consistently, Western blot analysis confirmed corresponding increases in ZO-1 and Occludin protein expression in the *W. cibaria* group (Fig. 2C). IHC of colon sections demonstrated pronounced upregulation of ZO-1 and Occludin protein expression in the *W. cibaria* group (Fig. 2D). Collectively, these findings demonstrate that *W. cibaria* supplementation

strengthens intestinal barrier function in CAC mice by upregulating tight junction components.

## *W. cibaria* activates FXR in the intestine

To elucidate the anti-tumor mechanisms of *W. cibaria*, we performed RNA sequencing on colon tumor tissues from CAC model mice. Principal component analysis (PCA) revealed distinct transcriptional profiles between Pbs and *W. cibaria* groups (Fig. S2A). In the Kyoto Encyclopedia of Genes and Genomes (KEGG) enrichment analysis, the top pathways significantly enriched were those associated with inflammation, drug metabolism, and chemical carcinogenesis. Among them, the bile secretion pathway enriched in the top few attracted our attention (Fig. 3A). Key BA transporters and receptors—FXR (upregulated) and organic solute transporter alpha/beta (OST α/β, upregulated)—and the apical sodium-dependent bile acid transporter (ASBT, downregulated) were enriched in the bile secretion pathway. Complementary gene ontology (GO) analysis further implicated *W. cibaria* in restoring BA homeostasis, epithelial barrier function, and inflammation resolution (Fig. 3B). Interestingly, when analyzing the two main bile acid receptors with a volcano plot, there was no difference in TGR5 expression between the two groups, FXR was markedly upregulated in the *W. cibaria* group (Fig. S2B). Gene set enrichment analysis (GSEA) confirmed significant enrichment of the colorectal cancer pathway (Fig. 3C), supporting the tumor-suppressive effects of *W. cibaria*. We further verified the activation of the FXR pathway with qPCR. Compared with the Pbs group, colonic FXR, small heterodimer partner (SHP), fibroblast growth factor 15 (FGF15), OST α, and OST β mRNA levels were significantly higher in the *W. cibaria* group. No significant changes in TGR5 mRNA levels were observed, and ASBT mRNA levels were significantly reduced in the *W. cibaria* group (Fig. 3D and E). Consistently, the protein expression of FXR was also significantly increased in the colon tissue of *W. cibaria* groups (Fig. 3F). These findings collectively demonstrate that *W. cibaria* supplementation activates the colonic FXR signaling axis, establishing its role in modulating BA metabolism and suppressing tumors in CAC mice.

## Reduced FXR expression in CRC with poor patient prognosis

In CRC patients, reduced FXR expression in tumor tissues correlates with poor clinical prognosis (24). To validate this association, we conducted survival analysis using the cBioPortal database, which revealed significantly prolonged overall survival in CRC patients with high FXR expression compared to those with low FXR levels (Fig. 4A). Tissue-specific dysregulation of FXR was further explored via the GEPIA2 database. Notably, FXR expression was markedly downregulated in colon adenocarcinoma tumor tissues vs adjacent normal tissues, whereas no significant difference was observed in rectal adenocarcinoma cohorts (Fig. 4B). To confirm these bioinformatics findings, we analyzed matched clinical CRC specimens (tumor vs normal tissues). Quantitative qPCR demonstrated significantly lower FXR mRNA levels in tumor tissues (Fig. 4C), paralleled by reduced FXR protein expression in tumors via Western blot (Fig. 4D). These multimodal data robustly establish FXR downregulation as a hallmark of colorectal tumorigenesis and a potential prognostic biomarker. While our database analyses included diverse CRC subtypes, experimental validation focused on the AOM/DSS-induced CAC model. Importantly, molecular profiling confirms that AOM/DSS tumors recapitulate key pathogenic features of human CRC, including FXR suppression patterns (25). This phenotypic conservation underscores the translational relevance of our findings, suggesting that FXR-targeted therapeutic strategies, as demonstrated in the CAC model, may benefit multiple CRC subtypes.

## *W. cibaria* activates the FXR pathway by regulating BA

As a nuclear receptor for BAs, FXR plays a pivotal role in suppressing colorectal tumor progression through its ligand-dependent activation (26). Given the regulation between BAs and FXR signaling, we characterized BA metabolic profiles in colon contents from

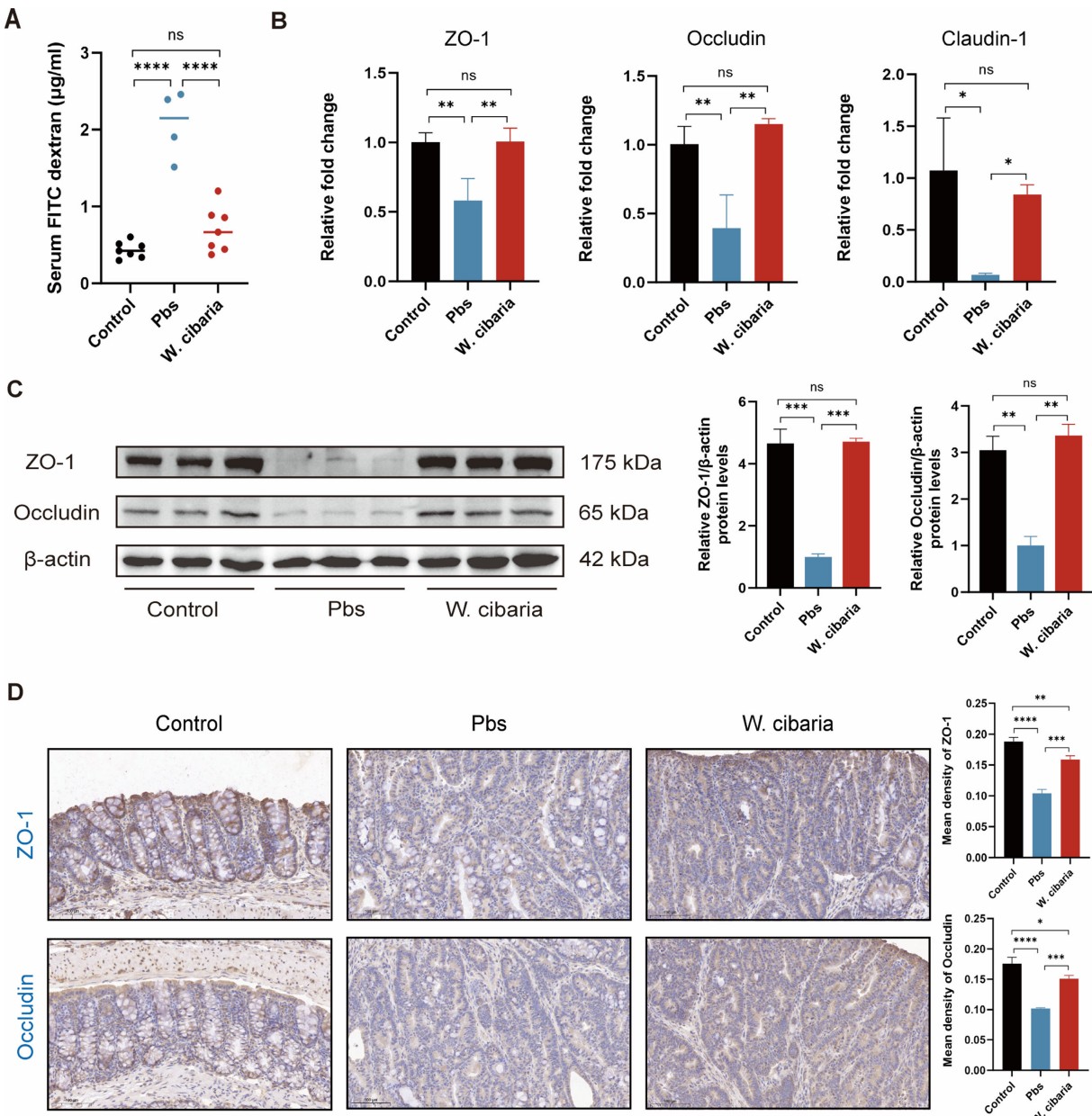

**FIG 2** *W. cibaria* restores the intestinal barrier in CAC mice. (A) FITC-dextran concentration (μg/mL) in mice serum . (B) The relative fold change in ZO-1, Occludin, and Claudin-1 expression by qPCR analysis ($n = 3$). (C) The protein expression of ZO-1 and Occludin in colon tissues. (D) Representative immunohistochemical plots and bar graphs of mouse intestinal tight junction proteins ZO-1 and Occludin for each group (scale bar = 100 μm). Data are expressed as the mean ± SEM. For comparisons between two groups, two-tailed unpaired Student's *t*-tests were used. For multi-group comparisons, one-way ANOVA followed by Tukey's *post hoc* test was applied. *$P < 0.05$; **$P < 0.01$; ***$P < 0.001$; ****$P < 0.0001$; ns, not significant.

experimental groups (27). PCA and partial least squares-discriminant analysis (PLS-DA) revealed distinct clustering patterns of BA compositions among Control, Pbs, and *W. cibaria* groups (Fig. 5A and B). The clustering heat map further identified differentially abundant BA species across groups (Fig. 5C). Notably, total BA levels and unconjugated BA levels were significantly reduced in the *W. cibaria* group compared to the Pbs group, reaching levels comparable to the Control group (Fig. 5D). There was no significant difference in conjugated BA levels (Fig. 5D) and the total abundance of conjugated or unconjugated BAs among the three groups (Fig. S3). Among the conjugated BAs, there were no different BAs among the three groups (Fig. 5E). But targeted quantification demonstrated a reduction of unconjugated BAs in the *W. cibaria* group—including DCA,

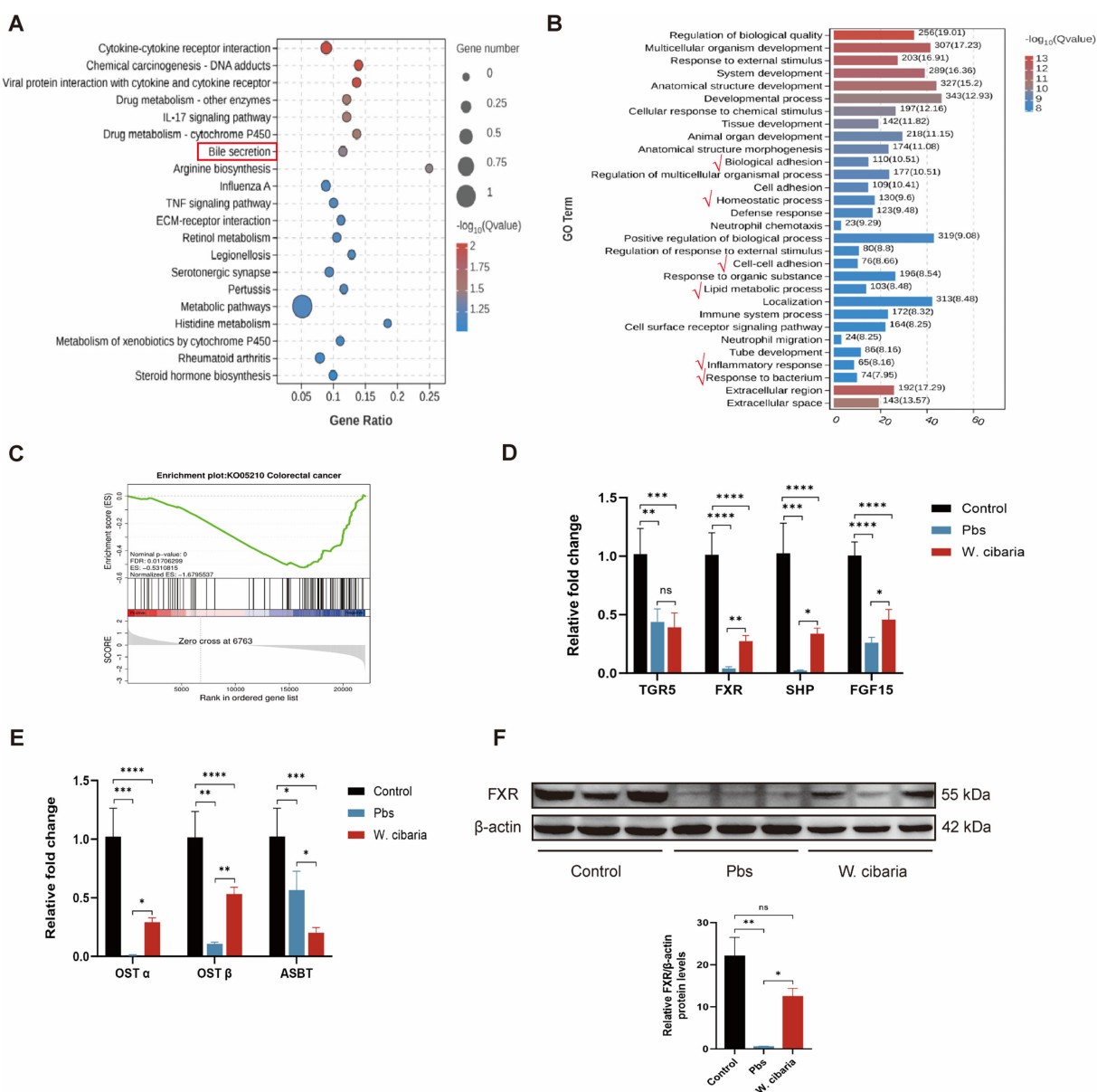

**FIG 3** *W. cibaria* activates FXR in the gut (*n* = 4). (A) Results of KEGG enrichment analysis of differential genes in RNA sequencing analysis (RNA-seq). (B) Results of GO enrichment analysis of differential genes in RNA-seq. (C) Results of GSEA enrichment analysis of the gene set of colorectal cancer. (D) The relative fold change in TGR5, FXR, SHP, and FGF15 expression by qPCR analysis. (E) The relative fold change in OST α, OST β, and ASBT expression by qPCR analysis. (F) The protein expression of FXR in colon tissues. Data are expressed as mean ± SEM. For multi-group comparisons, one-way ANOVA followed by Tukey's *post hoc* test was applied. *$P < 0.05$; **$P < 0.01$; ***$P < 0.001$; ****$P < 0.0001$; ns, not significant.

LCA, UCA, UDCA, βUDCA, and ωMCA—ultimately matching control group levels (Fig. 5F). This finding carries mechanistic significance, as DCA, LCA, and UDCA are established FXR antagonists that repress receptor activity (28–30). The *W. cibaria*-induced reduction of these inhibitory BAs creates a permissive microenvironment for FXR activation, consistent with our observed upregulation of FXR signaling. Collectively, these results demonstrate that *W. cibaria* supplementation normalizes BA dysmetabolism in CAC mice by selectively decreasing FXR-antagonistic unconjugated BAs, thereby promoting FXR activation.

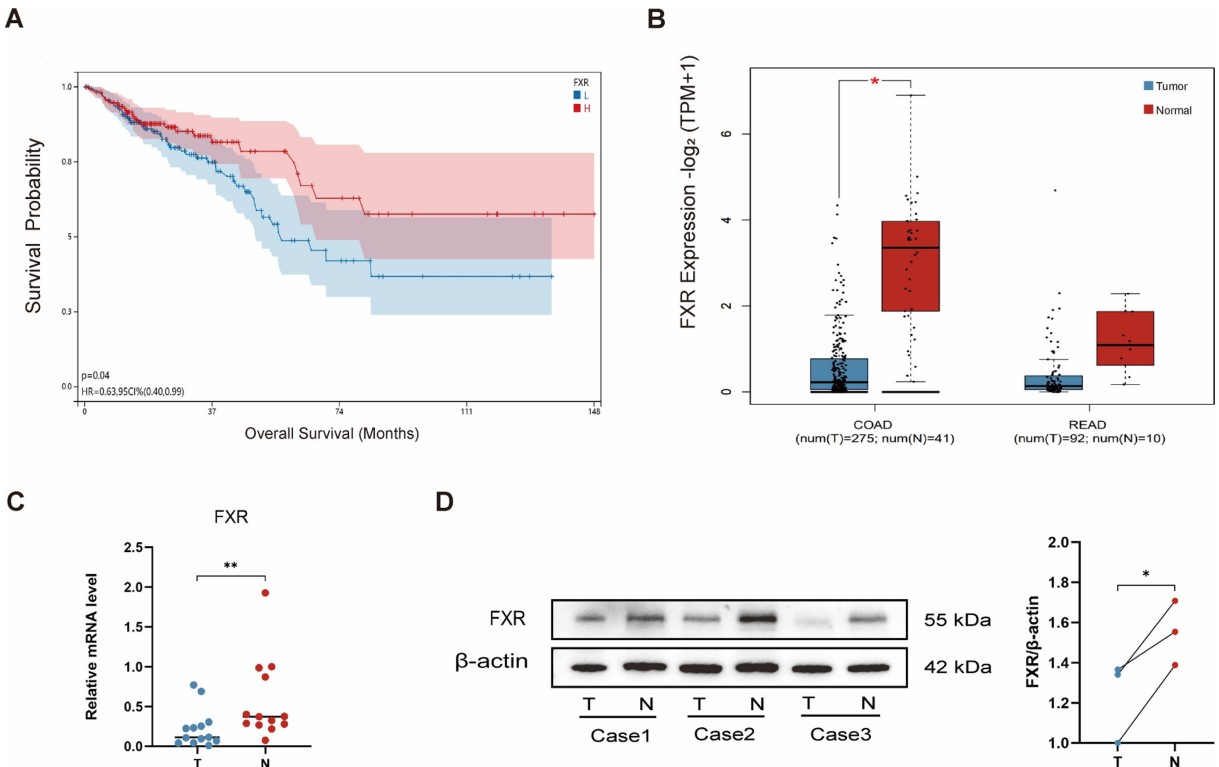

**FIG 4** Intestinal FXR expression profile is important in colorectal cancer. (A) The relationship between overall survival time and FXR expression in CRC patients. (B) Comparison of FXR expression in tumor tissue and normal tissue of patients with COAD and READ. Blue box plot represents T (tumor); red box plot represents N (normal). (C) Relative mRNA expression of FXR in tumor tissue (T) and normal tissue (N) of CRC patients ($n$ = 13). (D) The protein expression of FXR in tumor tissue (T) and normal tissue (N) of CRC patients. Data are expressed as the mean ± SEM. For comparisons between two groups, two-tailed unpaired Student's $t$-tests were used. *$P$ < 0.05; **$P$ < 0.01; ***$P$ < 0.001; ****$P$ < 0.0001; ns, not significant.

## *W. cibaria* affects the gut microbiota to influence BAs

Gut microbial communities play an essential role in mediating intestinal tumorigenesis (31). To investigate the potential involvement of *W. cibaria* in bile acid metabolism, we conducted 16S rRNA sequencing analysis of mouse colonic contents. While Chao, Simpson, and Shannon indices showed differences in α-diversity among the three groups (Fig. S4A), principal coordinate analysis demonstrated β-diversity differences in gut microbiota composition between groups (Fig. S4B). Considering that bile salt hydrolase mediates the deconjugation of conjugated BAs to form unconjugated BAs (32), and our previous findings of elevated unconjugated BAs in the colonic contents of the Pbs group, we hypothesized that increased BSH-producing bacteria might underlie this phenomenon. LEfSe identified significant enrichment of specific taxa in the Pbs group, including *Bacilli* (class), *Lactobacillales* (order), *Lactobacillus* (genus), *Sphingomonas* (genus), and *Lactobacillaceae* (family) (Fig. 6A). This study established evidence that *Lactobacillaceae* possesses BSH activity (9). The analysis showed changes in the abundance of the top 10 dominant microbiota species at the family level in each group (Fig. 6B). Subsequent family-level analysis revealed that the Pbs group showed the most pronounced differences in *Lactobacillaceae* abundance (Fig. S4C through G).

qPCR quantification confirmed successful intestinal colonization of *W. cibaria* (Fig. 6C). Correspondingly, we observed significantly reduced BSH activity in the colonic contents of the *W. cibaria* group compared to the Pbs group (Fig. 6D). Our results show that *Lactobacillaceae* abundance was positively correlated to the content of total unconjugated bile acids (Fig. 6E). These findings collectively suggest that *W. cibaria* administration modulates gut microbiota composition in CAC mice, particularly reducing BSH-producing *Lactobacillaceae* abundance and the production of total unconjugated bile acids.

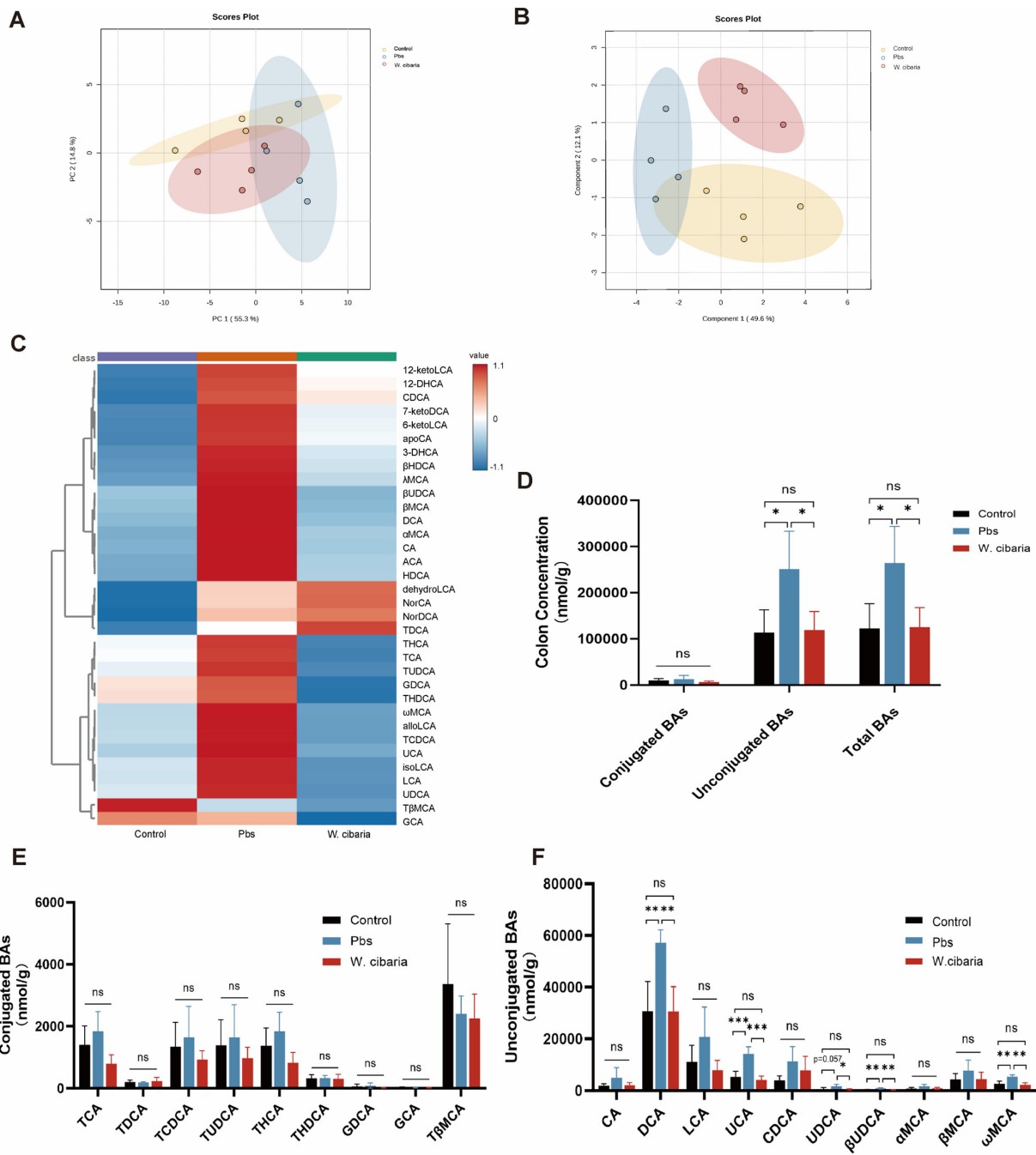

**FIG 5** *W. cibaria* activates FXR by altering bile acid composition. (A) PCA score plot of the colonic BA profiles. (B) PLSDA score plot of the colonic BA profiles. (C) Clustering heat map of BAs. (D) Quantification of BAs in the colonic contents. (E) Changes in the concentration of conjugated BAs in the colonic contents. (F) Changes in the concentration of unconjugated BAs in the colonic contents. Data are expressed as mean ± SEM. For multi-group comparisons, one-way ANOVA followed by Tukey's *post hoc* test was applied. *$P < 0.05$; **$P < 0.01$; ***$P < 0.001$; ****$P < 0.0001$; ns, not significant.

## FXR deficiency in the colon disrupts the protective role of *W. cibaria*

We established FXR^fl/fl and FXR^cKO mouse models of CAC (Fig. 7A). Strikingly, *W. cibaria* supplementation failed to reduce colon tumor numbers in FXR^cKO mice compared to FXR^fl/fl mice (Fig. 7B). Monitoring revealed no significant differences in percentage change of body weight or survival rates between *W. cibaria*-treated FXR^cKO mice and untreated FXR^cKO mice (Fig. 7C). Consistent with these observations, tumor burden metrics— including tumor number, size, and load—remained unaltered in FXR^cKO mice following *W.*

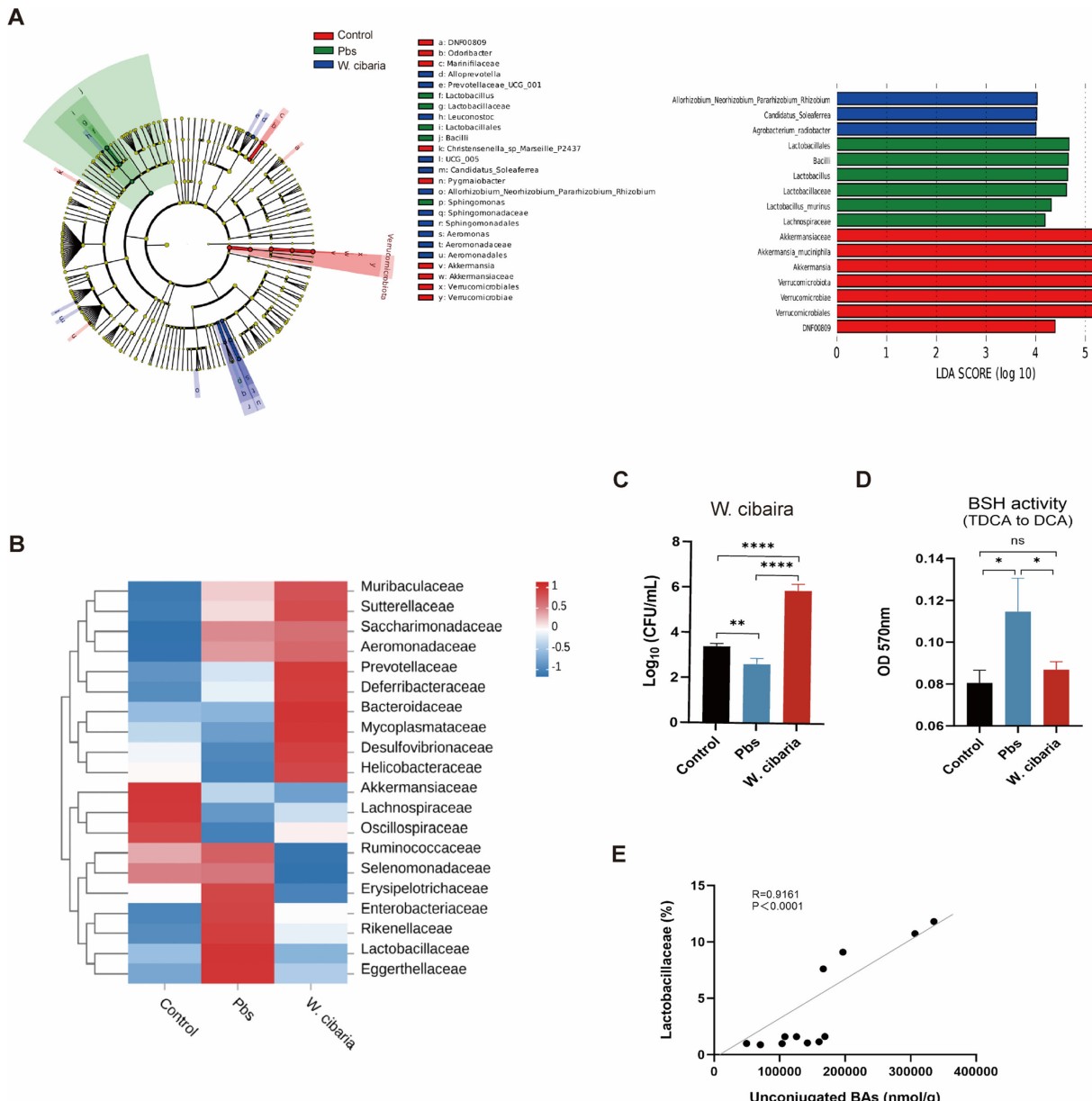

**FIG 6** *W. cibaria* alters the gut microbiota of CAC mice (*n* = 4). (A) LEfSe cladogram representing taxon enrichment and discriminative microbial biomarkers (LDA score of >3.5, *P* < 0.05). (B) Heatmap of abundance of the top 10 most abundant taxa at the family level. (C) Quantification of *W. cibaria* in colonic contents by qPCR. (D) Comparison of BSH enzyme activity in colonic contents. (E) The correlation between unconjugated BA level and *Lactobacillaceae* abundance was analyzed using Spearman's correlation. Data are expressed as the mean ± SEM. For multi-group comparisons, one-way ANOVA followed by Tukey's *post hoc* test was applied. *P < 0.05; **P < 0.01; ***P < 0.001; ****P < 0.0001; ns, not significant.

*cibaria* intervention, with persistent colon shortening, indicating unabated disease progression (Fig. 7D through G). Histopathological analysis via H&E staining further confirmed the lack of improvement in colonic dysplasia in FXR^cKO mice administered with *W. cibaria* (Fig. 7H). In stark contrast, FXR^fl/fl mice exhibited pronounced therapeutic responses to *W. cibaria*, demonstrating significant reductions in tumor incidence, attenuated body weight loss, and diminished tumor dimensions. These data collectively indicate that the antitumor efficacy of *W. cibaria* in CAC requires colonic FXR signaling. Furthermore, FXR deficiency in the colon appears to drive tumorigenesis, as evidenced by the exacerbated neoplastic phenotype in FXR^cKO mice regardless of *W. cibaria* treatment.

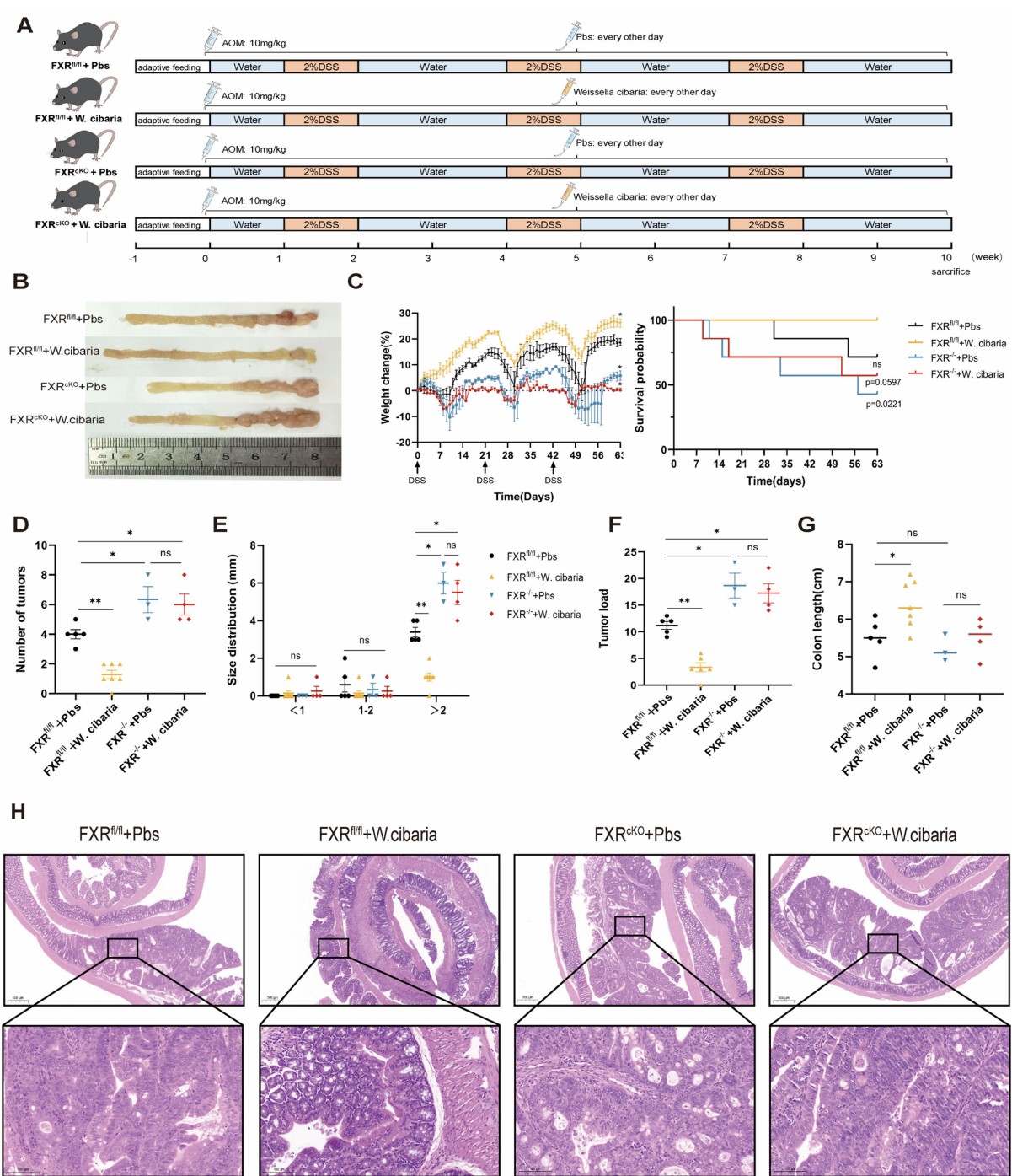

**FIG 7** FXR of the colon plays a crucial role in ameliorating tumorigenesis in CAC mice by *W. cibaria* . (A) Animal experiment protocol (*n* = 7). (B) Macroscopic view of the mouse colon. (C) Percentage change in body weight for each group of mice based on initial body weight (first feeding of 2% DSS water) and overall survival time of mice. (D through F) Tumor number, tumor size distribution, and tumor load in the colon of mice (tumor load values were calculated using the formulas provided in Materials and Methods and tumor size distribution data from panel E). (G) Intestinal length. (H) Representative mouse intestinal H&E staining images (scale bar = 500 and 100 μm). The asterisks or ns directly labeled on the scatterplot are significance analyses of differences between the latter groups compared with the first group (FXR$^{fl/fl}$ + Pbs group). Data are expressed as mean ± SEM. For multi-group comparisons, one-way ANOVA followed by Tukey's *post hoc* test was applied. *$P < 0.05$; **$P < 0.01$; ***$P < 0.001$; ****$P < 0.0001$; ns, not significant.

## W. cibaria activates FXR to suppress NF-κB and reduce inflammatory cytokine release

The study demonstrated that activation of FXR in inflammatory bowel disease inhibits the NF-κB pathway in the gut (33). To investigate the mechanistic role of the FXR-NF-κB pathway in CAC, we analyzed p65 and phosphorylated-p65 (p-p65) expression in colon tissues by Western blotting. Total p65 levels remained consistent across all experimental groups. However, phosphorylation status diverged: supplementation with *W. cibaria* significantly suppressed p-p65 in FXR^fl/fl mice compared to their Pbs-treated counterparts, whereas no such effect was observed in FXR^cKO mice. Strikingly, both FXR^cKO + Pbs and FXR^cKO + *W. cibaria* groups demonstrated robust p-p65 upregulation relative to the FXR^fl/fl+ Pbs group (Fig. 8A).

We next assessed inflammatory cytokine levels in mouse colon tissues via ELISA. Comparative analysis revealed no significant differences in IL-1β, IL-6, and TNF-α expression between the FXR^cKO + *W. cibaria* and FXR^cKO + Pbs groups. In contrast, mice from the FXR^fl/fl + *W. cibaria* group exhibited markedly reduced levels of all three cytokines relative to the FXR^fl/fl + Pbs group. Notably, both FXR^cKO + Pbs and FXR^cKO + *W. cibaria* groups displayed significantly elevated IL-6 expression compared to the FXR^fl/fl+ Pbs group (Fig. 8B). These findings suggest that the ability of *W. cibaria* to inhibit colorectal tumorigenesis is achieved through the activation of FXR in the intestine, inhibiting the NF-κB pathway by inhibiting p65 phosphorylation and reducing the release of inflammatory factors.

## DISCUSSION

In this study, we elucidated the inhibitory role of *W. cibaria* in colorectal tumorigenesis. In CAC mice, *W. cibaria* treatment induced a functional partial shift in the gut microbiota, characterized by a reduction in BSH enzyme-producing bacteria. This shift was associated with improved bile acid homeostasis, activation of FXR, and suppression of colorectal tumorigenesis.

While multiple therapeutic strategies exist for colorectal cancer, probiotics have emerged as promising adjuvant agents due to their dual preventive and therapeutic potential (34). Among probiotic candidates, the genus *Weissella* has garnered increasing attention for its versatile applications in food technology and biomedical sciences (35). Previous studies have demonstrated *W. cibaria*'s ability to improve oral health through gingival microbiota modulation (36) and enhance immune function via NK cell activation (37). However, its *in vivo* anticancer efficacy remained unexplored. Our study pioneers the use of *W. cibaria* in an AOM/DSS-induced CAC model, systematically evaluating its cumulative effects on microbiota modulation, inflammatory suppression, and tumor prevention throughout carcinogenesis.

The potential mechanism underlying *W. cibaria*'s antitumor effects was revealed through KEGG pathway analysis, which identified BA secretion as a significantly enriched pathway. Although the canonical BA secretion pathway is annotated with hepatic functions, its enrichment in colon tumors primarily reflects dysregulation of intestinal bile acid reabsorption and receptor signaling, rather than secretion following *de novo* synthesis. Specifically, the upregulation of FXR and OSTα/OSTβ, coupled with downregulation of ASBT, suggests that tumor-specific adaptations under the influence of *W. cibaria* modulate luminal bile acid levels. This finding aligns with our observation of upregulated FXR expression in colonic tumors, corroborating previous reports that FXR activation inhibits CRC progression and represents a promising therapeutic target (38). Targeted BA profiling demonstrated that *W. cibaria* supplementation reverses CAC-associated BA dysregulation—notably reducing DCA, LCA, and UDCA levels. Crucially, our experiments in intestinal FXR-deficient mice confirmed the indispensability of FXR signaling for *W. cibaria*'s antitumor efficacy, establishing BA profile correction as a key mechanism.

The dual enzymatic activity of BSH—mediating both deconjugation of primary BAs and conjugation of amino acids to CA/CDCA to produce microbially conjugated bile

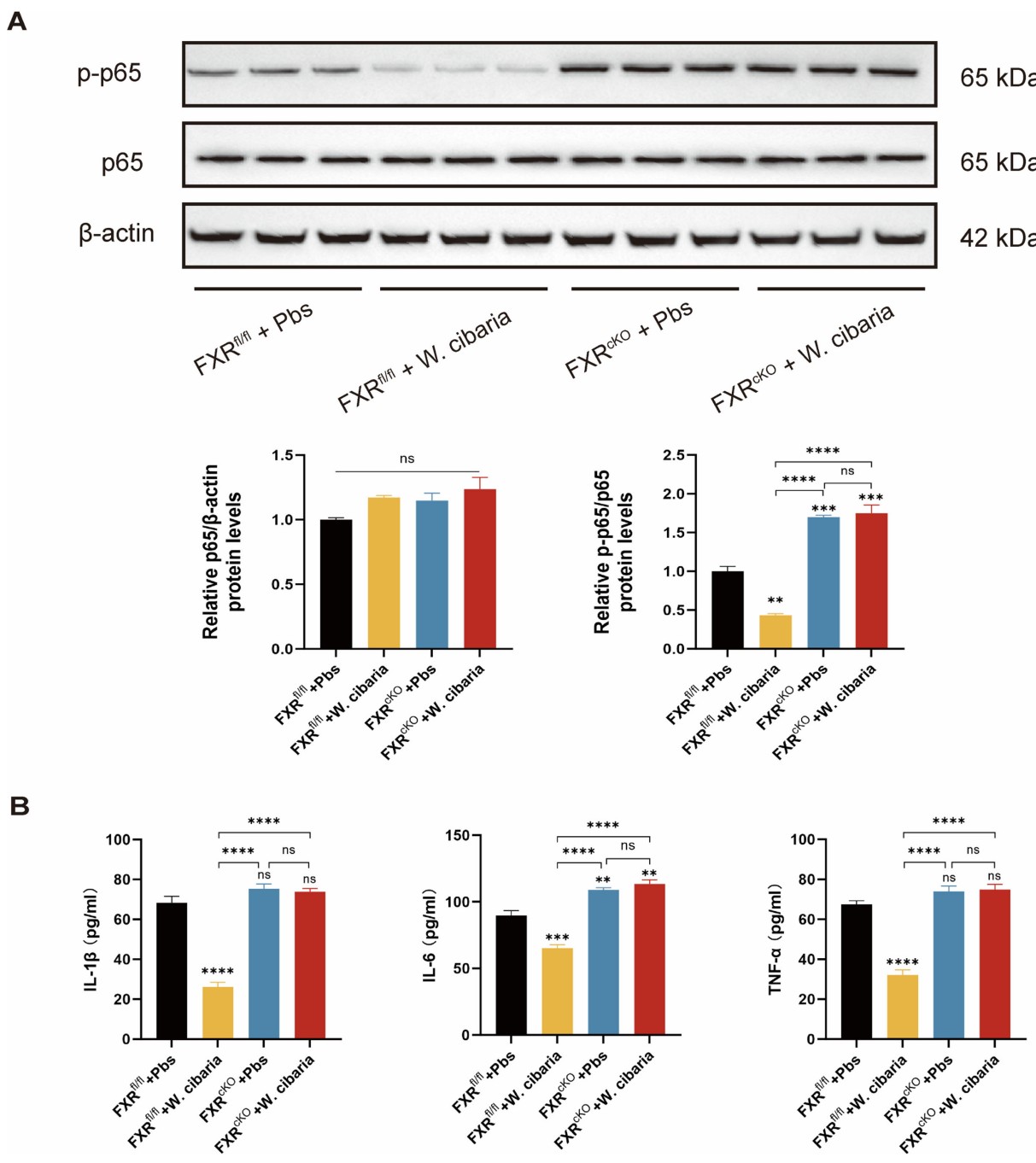

FIG 8 The important role of FXR in NF-κB and inflammatory factors in colon tissues. (A) The protein expression of p-p65 and p65 in colon tissues. (B) ELISA was performed to measure IL-1β, IL-6, and TNF-α expression levels in colon tissues. The asterisks or ns labeled directly on the bar charts are the significance analyses of the differences between the latter groups compared to the first group (FXR^fl/fl + Pbs group). Data are expressed as the mean ± SEM. For multi-group comparisons, one-way ANOVA followed by Tukey's *post hoc* test was applied. *$P < 0.05$; **$P < 0.01$; ***$P < 0.001$; ****$P < 0.0001$; ns, not significant.

acids (MCBAs)—underscores its complex role in BA metabolism (39). Our findings revealed that *W. cibaria* treatment significantly reduces intestinal unconjugated BAs, potentially through selective depletion of BSH-active *Lactobacillaceae* (9). The observed positive correlation between *Lactobacillaceae* abundance and unconjugated BA levels suggests that *W. cibaria* modulates BA metabolism through targeted microbial remodeling. Importantly, while conjugated BA levels remained unchanged, the significant reduction in unconjugated BAs highlights the specificity of this microbial-BA interaction.

NF-κB, a master regulator of inflammatory responses, was identified as a critical downstream effector in our mechanistic framework. The present study demonstrates that CAC-induced NF-κB pathway activation and subsequent inflammatory factor release can be effectively suppressed by *W. cibaria* supplementation. This observation extends previous findings of FXR-NF-κB cross talk in hepatocarcinogenesis and metabolic disorders (38–41) to the CRC context. Our data suggest a novel regulatory axis where FXR deficiency promotes tumorigenesis through NF-κB hyperactivation, while *W. cibaria* intervention restores homeostatic balance through BA-mediated FXR activation. Our data suggest a novel regulatory axis where FXR deficiency promotes tumorigenesis through NF-κB hyperactivation, while *W. cibaria* intervention restores homeostatic balance through BA-mediated FXR activation. The mechanistic insights gained from this study could provide a basis for the development of targeted therapeutic interventions.

This investigation provides the first *in vivo* evidence of *W. cibaria*'s antitumor efficacy in CRC, establishing crucial theoretical foundations for its clinical translation. However, two key limitations warrant consideration: first, our study did not address the relationship between BSH activity and MCBA diversity—a critical knowledge gap requiring comprehensive metabolomic profiling in future research. Second, the paradoxical observation of decreased unconjugated BAs without concomitant changes in conjugated BAs merits mechanistic exploration. Subsequent studies should employ isotopic tracing and microbial genetic manipulation to dissect these complex BA dynamics. Finally, the downstream mechanisms associated with FXR are worth exploring further.

Our findings establish that *W. cibaria* inhibits CAC occurrence through coordinated modulation of the gut microbiota-BA-FXR axis. Specifically, *W. cibaria* can reduce BSH-active *Lactobacillaceae* abundance, decrease unconjugated BA levels, activate intestinal FXR signaling, and suppress NF-κB-mediated inflammation. These mechanistic insights position *W. cibaria* as a promising probiotic candidate for CRC prevention and adjuvant therapy, warranting further clinical investigation.

## ACKNOWLEDGMENTS

This work was supported by the National Natural Science Foundation of China (No. 82060440), the Incubation Project of the National Natural Science Foundation of China at Guiyang Medical University Affiliated Hospital (No. gyfynsfc [2022]-5), and the Guizhou Provincial Basic Research Program (No. Qiankehe Foundation -zk [2023] Important 043).

## AUTHOR AFFILIATIONS

[1]Department of Colorectal Surgery, Affiliated Hospital of Guizhou Medical University, Guiyang, Guizhou, China
[2]Guizhou Medical University, Guiyang, Guizhou, China
[3]Department of Oncology, Affiliated Hospital of Guizhou Medical University, Guiyang, Guizhou, China
[4]Department of Gastroenterology, Affiliated Hospital of Guizhou Medical University, Guiyang, Guizhou, China

## AUTHOR ORCIDs

Qiuyao Hao http://orcid.org/0009-0009-3910-3094
Weiwei Chen http://orcid.org/0000-0002-8275-9162
Yunhuan Zhen http://orcid.org/0000-0003-0809-0621

## AUTHOR CONTRIBUTIONS

Qiuyao Hao, Conceptualization, Data curation, Investigation, Methodology, Project administration, Validation, Writing – original draft | Fei Huang, Formal analysis, Software, Validation, Writing – review and editing | Liangzheng Chang, Formal analysis,

Methodology, Visualization | Hongyuan Dai, Formal analysis, Software, Visualization | Weiwei Chen, Funding acquisition, Project administration, Supervision, Writing – review and editing | Yiran Yao, Data curation, Resources, Visualization, Writing – review and editing | Yunhuan Zhen, Funding acquisition, Resources, Supervision, Writing – review and editing

## ETHICS APPROVAL

The related human studies were approved by the Ethics Committee of Affiliated Hospital of Guizhou Medical University (approval number: 2020-030). The patients provided their written informed consent to participate in this study. All animal experiments were conducted in accordance with the ethical requirements of the Guizhou Medical University Animal Care Welfare Committee (approval number: 2000646).

## ADDITIONAL FILES

The following material is available online.

### Supplemental Material

**Fig. S1 (mSystems00288-25-s0001.tif).** Protein expression of FXR in colon tissues.
**Fig. S2 (mSystems00288-25-s0002.tif).** PCA analysis for RNA-seq detection of colon tumor and differential gene volcano map.
**Fig. S3 (mSystems00288-25-s0003.tif).** Relative abundance ratios of various types of bile acids.
**Fig. S4 (mSystems00288-25-s0004.tif).** α-Diversity indexes, PCoA, and comparative abundance of the top 5 most abundant taxa at the family level.
**Legends (mSystems00288-25-s0005.docx).** Legends for Figures S1 to S4.
**Table S1 (mSystems00288-25-s0006.pdf).** Primers for mouse identification.
**Table S2 (mSystems00288-25-s0007.pdf).** Species-specific primers and probes used for qPCR.
**Table S3 (mSystems00288-25-s0008.pdf).** Full names of the bile acids in Fig. 6.
**Table S4 (mSystems00288-25-s0009.pdf).** Primers for real-time PCR.

### Open Peer Review

**PEER REVIEW HISTORY (review-history.pdf).** An accounting of the reviewer comments and feedback.

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
