## [Reviewer comments · mSystems]

Weissella cibaria Suppresses Colitis-Associated Colorectal Cancer by Modulating the Gut Microbiota–Bile Acid–FXR Axis

qiuyao hao, fei huang, Liangzheng Chang, hongyuan dai, weiwei chen, yiran yao, and Yunhuan Zhen

Corresponding Author(s): qiuyao hao, Guizhou Medical University

Review Timeline:

Submission Date:	March 5, 2025
Editorial Decision:	April 10, 2025
Revision Received:	May 8, 2025
Accepted:	May 16, 2025

Editor: Neha Garg

Reviewer(s): Disclosure of reviewer identity is with reference to reviewer comments included in decision letter(s). The following individuals involved in review of your submission have agreed to reveal their identity: Ipsita Mohanty (Reviewer #2)

Transaction Report:

DOI: <https://doi.org/10.1128/msystems.00288-25>

Re: mSystems00288-25 (Weissella cibaria Suppresses Colitis-Associated Colorectal Cancer by Modulating the Gut Microbiota-Bile Acid-FXR Axis)

Dear Dr. Qiuyao hao:

Thank you for the privilege of reviewing your work. Below you will find instructions from the mSystems editorial office, and the reviewer comments. Based on reviewer 1 comments, the the decision is to revise the manuscript. The concerns of reviewer 1 need to be carefully addressed before we can accept the manuscript.

Revision Guidelines

Sincerely,
Neha Garg
Editor
mSystems

Reviewer #1 (Comments for the Author):

This is a revised manuscript. In this study, Hao et al. investigate the anti-inflammatory and anti-tumor effects of the bacterial strain *Weissella cibaria* in an AOM/DSS mouse model. My detailed questions on revision see below:

Comments

1. "In Figure 1, by the end of inoculation, did the author measure the *W. cibaria* final levels in the gut? "----How to convert qPCR quantification results of *W. cibaria* to CFU/ml?
2. "In Figure 1G, what is tumor load? Tumor volume? Units need to be labeled." ----Based on what author described, the the y-axis should indicate the number and the x-axis the size distribution, based on the data presented
3. In Figure 2c, the westernblot of ZO-1, why there is no band at all in pbs group?
In Figure 4C, the author rewrite it "To enhance clarity, we have added the following statement in the revised Results section:"
Key BA transporters and receptors - including FXR, ASBT, OST α /OST β - were significantly enriched in the bile secretion pathway." ----Our common knowledge is that bile is secreted by the liver, not the colon. I have serious doubts about the authors' sequencing results in the colonic tissue samples, that all these BAs influx and efflux transporters are highly expressed.
4. In Figure 7, what is pp65? There are too many inconsistencies in the writing, such as the unclear use of asterisks in comparison groups.

Reviewer #2 (Comments for the Author):

Thank you for addressing the comments.

The authors have addressed the concerns in the revision.

Thank you for your continued guidance and the opportunity to refine our manuscript further. We sincerely appreciate the reviewers' constructive feedback, which has helped us strengthen key aspects of this work. We have carefully addressed all remaining comments as outlined below, with corresponding revisions highlighted in [red color] in the manuscript. A final proofreading has been conducted to ensure consistency throughout the paper.

Response to the Reviewer #1:

Comment 1: "In Figure1, by the end of inoculation, did the author measure the *W. cibaria* final levels in the gut? "----How to convert qPCR quantification results of *W.cibaria* to CFU/ml?

Response:

We sincerely thank the reviewers for their insightful questions regarding the quantification of *W. cibaria* colonization. We recognize that the methodology for converting qPCR data to CFU/ml was not sufficiently detailed in the original manuscript. Below, our clarifications are described and added in the Methods section of the article:

1. Quantification of *W. cibaria* in the colonic contents of various groups of mice

Experimental procedure:(1) Sample collection. (2) DNA extraction. (3) qPCR analysis.

2. qPCR data were converted to CFU/ml, and a standard curve was generated

(1) Gradient concentration *W. cibaria* bacterial suspension preparation: serial dilution of bacterial suspension in PBS into 7 concentration gradients of 10^2 CFU/ml, 10^3 CFU/ml, 10^4 CFU/ml, 10^5 CFU/ml, 10^6 CFU/ml, 10^7 CFU/ml and 10^8 CFU/ml.

(2) DNA extraction: bacterial DNA was extracted from gradient dilutions of *W. cibaria* (10^2 - 10^8 CFU/ml) using the Bacterial Genomic DNA Extraction Kit (DP302, Tiangen, Beijing, China).

(3) qPCR was performed under the same conditions as the experimental samples.

(4) Production of standard curve: threshold cycle (Ct) values were plotted as a standard curve based on \log_{10} transformed CFU/ml.

(5) Calculation of experimental samples: the Ct values of colon contents samples were interpolated into the standard curve equation to obtain \log_{10} (CFU/ml), and the inverse logarithmic transformation was performed to obtain the final CFU/ml values.

3. Methodological corrections

We will add a supplementary description of the conversion of qPCR data to CFU/ml in the Methods section.

Revision: Methods section. "A real-time quantitative polymerase chain reaction (RT-qPCR) technique was used to determine the colonization level of *W. cibaria* in the mouse colon. To establish a standard curve, serial dilutions of leech suspensions were made in sterile PBS to a final concentration of 10^2 - 10^8 CFU/mL, and genomic DNA was extracted from each dilution using a bacterial genomic DNA extraction kit (DP302, Tiangen, China, Beijing). For experimental samples, a QIAamp Fast DNA Stool Mini Kit (Qiagen, Hilden, Germany) was used to isolate DNA from the colon contents of each group of mice. qPCR was performed on a QuantStudio 6 Real-Time PCR system (Thermo Fisher Scientific) using the Taq Pro U+ Multiple Probe qPCR Mix (QN213-01, Vazyme, Nanjing, China). The primers and double-labeled probe sequences used are shown in Table S2. A standard curve was made by plotting threshold cycle (CT) values against \log_{10} -transformed CFU/mL counts (*W. cibaria*: 10^2 - 10^8 CFU/mL). The CT values of the colon contents of each group were plugged into this equation for CFU/ml transformation to obtain the level of *W. cibaria* colonization in each group (18,19)." (Lines 157-169)

Comment 2: "In Figure 1G, what is tumor load? Tumor volume? Units need to be labeled." ----Based on what author described, the the y-axis should indicate the number and the x-axis the size distribution, based on the data presented

Response:

We thank the reviewers for their attention to methodological clarity. We sincerely apologise for the remaining ambiguity and have implemented the following revisions to address the issue fully.

1. Conceptual difference between “tumor load” and “tumor volume”: (1) As highlighted in the Methods section of the article, tumor load is a dimensionless composite score reflecting the distribution of tumor numbers and sizes. It is calculated as follows: Tumor load = $(N_1 \times 1) + (N_2 \times 2) + (N_3 \times 3)$, where N_1 , N_2 , and N_3 represent the number of small (< 1 mm), medium (1- 2 mm), and large (> 2 mm) tumors, respectively. (2) Tumor volumes were not analysed in this study as described in the revised text (3D measurements are required).

In the text, we mentioned tumor load, but did not perform tumor volume counts for the time being. We have further added explanations in the Methods section and on the figure notes of Figure 1 and Figure 7, which hopefully address the comment you raised.

Revision: Methods section: “Tumor load represents a dimensionless composite metric derived from tumor number and size stratification, where elevated scores correlate with increased neoplastic burden” (Lines 113-115)

Figure 1: “(E-G) Tumor number, tumor size distribution, and tumor load in the colon of mice (Tumor load values were calculated using the formulas provided in the methods section and tumor size distribution data from Fig. 1F).” (Lines 625-627)

Figure 7: “(D-F) Tumor number, tumor size distribution, and tumor load in the colon of mice (Tumor load values were calculated using the formulas provided in the methods section and tumor size distribution data from Fig. 7E).” (Lines 676-679)

2. In colorectal cancer-related research articles, the method of calculating tumor load varies from study to study; however, tumor load itself is a unitless parameter under this calculation.

(Article 1) Method: According to the diameter of the tumors in mice colon on day 90 of AOM/DSS model, we divided them into three group: small tumors, <1 mm; medium tumors, $1 \text{ mm} \leq$ and $\leq 2 \text{ mm}$; large tumors, $> 2 \text{ mm}$. Tumor load was calculated according to the following formula: tumor load = (number of small tumors) \times 1 + (number of medium tumors) \times 2 + (number of large tumors) \times 3 [1].

Figure on tumor load:

(Article 2) Method: the tumor size distribution according to tumor numbers (tumor load) were recorded [2]

Figure on tumor load:

Reference:

[1] Tian, M., Wang, X., Sun, J., Lin, W., Chen, L., Liu, S., Wu, X., Shi, L., Xu, P., Cai, X., & Wang, X. (2020). IRF3 prevents colorectal tumorigenesis via inhibiting the nuclear translocation of β -catenin. *Nature communications*, 11(1), 5762.

<https://doi.org/10.1038/s41467-020-19627-7>

[2] Tan, G., Huang, C., Chen, J. et al. HMGB1 released from GSDME-mediated pyroptotic epithelial cells participates in the tumorigenesis of colitis-associated colorectal cancer through the ERK1/2 pathway. *J Hematol Oncol* 13, 149 (2020).

<https://doi.org/10.1186/s13045-020-00985-0>

Comment 3: In Figure 2c, the westernblot of ZO-1, why there is no band at all in pbs group?

Response:

We thank the reviewer for raising this critical concern. The absence of visible ZO-1 bands in the PBS control group of the original Western blot (Figure 2C) is primarily due to low protein abundance combined with suboptimal imaging parameters. To address this issue, we provide the following clarifications and validation data:

1. Image Acquisition Limitations: Possibly due to overexposure of the high-abundance bands in the *W. cibaria* and Control groups, the low exposure time/gain settings of the raw images inadvertently masked the weak ZO-1 signal in the Pbs group.

2. Raw data are attached: We provide annotated raw blots showing faint ZO-1 bands (red arrows) in Pbs group staining lanes. However, for visual clarity, we have reduced the display of these bands in the figure after adjusting the contrast in post-mapping.

3. To address potential variability, we repeated the Western Blot under identical conditions. The repeated results showed marginally improved visibility of ZO-1 in the PBS group, while the overall expression trends across all groups remained consistent with the original data. This reinforces the biological relevance of the observed differences.

Revision: In response to reviewer feedback, the original Western blot image in Figure 2C has been replaced with a replicate experiment conducted under adjusted exposure conditions to ensure accurate representation of ZO-1 expression levels in the PBS control group. The revised image confirms the original findings, as quantified in the new Figure 2C.

Comment 4: In Figure 4C, the author rewrite it "To enhance clarity, we have added the following statement in the revised Results section:" Key BA transporters and receptors - including FXR, ASBT, OST α /OST β - were significantly enriched in the bile secretion pathway." ----Our common knowledge is that bile is secreted by the liver, not the colon. I have serious doubts about the authors' sequencing results in the colonic tissue samples, that all these BAs influx and efflux transporters are highly expressed.

Response:

We sincerely appreciate the reviewer's critical comment regarding the enrichment of bile acid (BA)-related transporters and receptors in colon tumors. We acknowledge that the liver is the primary site of BA synthesis and secretion, and under normal physiological conditions, the colon does not actively secrete bile acids.

1. FXR, ASBT, and OST α /OST β act as typical bile acid influx and efflux transporters. FXR is highly expressed in normal liver [1]; ASBT and OST α /OST β are lowly expressed in liver, among which OST α /OST β can be highly expressed in pathological states of the liver [2]. Whereas FXR, ASBT, and OST α /OST β are expressed in the normal colon [2-3].
2. The supplementary note we added last time in the results section still has an unclear presentation. The section "FXR, ASBT, OST α /OST β - Significantly enriched in the bile secretion pathway" refers to the fact that these genes are functionally centered on a particular bile secretion pathway, rather than how the expression of individual genes varies. (1) We will revise the description of the results section again to clarify that the results are up-regulated for FXR and OST α /OST β and down-regulated for ASBT in the bile acid secretion pathway enriched by KEGG analysis. (2) OST α /OST β (BAs efflux transporters) and ASBT (BAs influx transporters) are not all highly expressed.

Revision: "Key BA transporters and receptors—FXR (upregulated) and organic solute transporter alpha/beta (OST α /OST β , upregulated)—and the apical sodium-dependent bile acid transporter (ASBT, downregulated) were enriched in the bile secretion pathway." (Lines 311-314)

3. The enrichment of the KEGG 'bile acid secretion' pathway in colon tumors might initially appear paradoxical, given that bile acid secretion is a hepatobiliary function unrelated to normal colonic physiology. However, this observation primarily arises from inherent limitations in the KEGG pathway annotation framework, which aggregates genes with pleiotropic roles under liver-centric terminology. To address this ambiguity, we have revised the Discussion section to explicitly state that the term 'bile secretion' in KEGG annotations should be interpreted as reflecting bile acid-associated signaling and metabolic reprogramming in tumors, rather than implying active bile acid secretion by

colonic tissues.

Revision: "The potential mechanism underlying *W. cibaria*'s antitumor effects was revealed through KEGG pathway analysis, which identified BA secretion as a significantly enriched pathway. Although the canonical BA secretion pathway is annotated with hepatic functions, its enrichment in colon tumors primarily reflects dysregulation of intestinal bile acid reabsorption and receptor signaling, rather than secretion following de novo synthesis. Specifically, the upregulation of FXR and OST α /OST β , coupled with downregulation of ASBT, suggests that tumor-specific adaptations under the influence of *W. cibaria* modulate luminal bile acid levels." (Lines 442-448)

4. We fully understand your concerns. However, in this study, we did send out colon tissue samples for examination. We also entrusted a company with professional qualifications to conduct the quality examination, testing and analysis of the samples, thereby ensuring the accuracy of the results. We also appreciate your valuable suggestions. In our future research, perhaps we can further explore the functions of *W. cibaria* in biological processes such as bile acid synthesis and secretion in the liver.

Reference:

[1] Luciano Adorini, Michael Trauner, FXR agonists in NASH treatment, *Journal of Hepatology*, Volume 79, Issue 5, 2023, Pages 1317-1331, ISSN 0168-8278, <https://doi.org/10.1016/j.jhep.2023.07.034>.

[2] Simbrunner B, Trauner M, Reiberger T. Review article: therapeutic aspects of bile acid signalling in the gut-liver axis. *Aliment Pharmacol Ther*. 2021; 54: 1243–1262. <https://doi.org/10.1111/apt.16602>

[3] Ling Xiao, Guoyu Pan, An important intestinal transporter that regulates the enterohepatic circulation of bile acids and cholesterol homeostasis: The apical sodium-dependent bile acid transporter (SLC10A2/ASBT), *Clinics and Research in Hepatology and Gastroenterology*, Volume 41, Issue 5, 2017, Pages 509-515, ISSN 2210-7401, <https://doi.org/10.1016/j.clinre.2017.02.001>.

Comment 5: In Figure 7, what is pp65? There are too many inconsistencies in the writing, such as the unclear use of asterisks in comparison groups.

Response:

Thank you for your careful review of this article and valuable comments! The issues you have pointed out are very critical, and we have reflected on them carefully and made targeted changes.

1. Regarding the definition of pp65 in **Figure 8**

We apologise for the oversight of not clearly explaining pp65 in the original article.

The abbreviation of phosphoinositide-p65 in the original article was inconsistent, so we have revised the abbreviation "pp65" to "p-p65 (phosphoinositide-p65)" in the article and the figure after referring to the high-scoring article [1,2].

The expression of p65 and phosphoinositide-p65 (p-p65) are key mediators of NF- κ B activation, and the status of p-p65 reflects the activation level of the pathway [3,4]. By detecting the expression of p65 and p-p65, this study aimed to investigate the association between FXR and NF- κ B pathway activity in CAC.

Revision: "The study demonstrated that activation of FXR in inflammatory bowel disease inhibits the NF- κ B pathway in the gut. To investigate the mechanistic role of the FXR-NF- κ B pathway in CAC, we analyzed p65 and **phosphorylated-p65 (p-p65)** expression in colon tissues by Western blotting. Total p65 levels remained consistent across all experimental groups. However, phosphorylation status diverged: supplementation with *W. cibaria* significantly suppressed **p-p65** in FXR^{fl/fl} mice compared to their PBS-treated counterparts, whereas no such effect was observed in FXR^{CKO} mice. Strikingly, both FXR^{CKO}+Pbs and FXR^{CKO}+*W. cibaria* groups demonstrated robust **p-p65** upregulation relative to the FXR^{fl/fl}+Pbs group (Fig. 8A)." **(Lines 410-417)**

Reference:

[1] Mi, Y., Mu, L., Huang, K. et al. Hypoxic colorectal cancer cells promote metastasis of normoxic cancer cells depending on IL-8/p65 signaling pathway. *Cell Death Dis* 11, 610 (2020). <https://doi.org/10.1038/s41419-020-02797-z>

[2] Shen, Z., Feng, X., Fang, Y. et al. POTE drives colorectal cancer development via regulating SPHK1/p65 signaling. *Cell Death Dis* 10, 863 (2019). <https://doi.org/10.1038/s41419-019-2046-7>

[3] Gutierrez, Humberto et al. Regulation of neural process growth, elaboration and structural plasticity by NF- κ B. *Trends in Neurosciences*, Volume 34, Issue 6, 316 - 325

[4] WANG Y, XIANG G-S, KOUROUMA F, et al. *Citrobacter rodentium*-induced NF- κ B

activation in hyperproliferating colonic epithelia: role of p65 (Ser536) phosphorylation [J]. British Journal of Pharmacology, 2006, 148(6): 814-24.

2. On inconsistencies in writing (e.g., unclear asterisk labelling of comparative groups)

Thank you for pointing out this omission. We have fully checked the full chart and have made revisions.

Correction of unclear comparison group labelling: The group comparison relationships in **Figure 7 and Figure 8** have been textually labelled in the figure notes to ensure that they can be understood visually by the reader.

Revision: "The asterisks or ns directly labelled on the scatterplot are significance analyses of differences between the later groups compared with the first group (FXR^{fl/fl}+Pbs group)." (Lines 680-682)

"The asterisks or ns labelled directly on the bar charts are the significance analyses of the differences between the later groups compared to the first group (FXR^{fl/fl}+Pbs group)." (Lines 687-689)

Thank you once again for your time and insightful comments. We believe that addressing these concerns has significantly improved the quality and clarity of our manuscript. We have carefully revised the text and figures in accordance with your suggestions. Please find our point-by-point responses to all comments above, and do not hesitate to contact us if further clarification or modifications are needed. We look forward to your feedback and hope that the revised manuscript now meets the journal's standards for publication.

Re: mSystems00288-25R1 (Weissella cibaria Suppresses Colitis-Associated Colorectal Cancer by Modulating the Gut Microbiota-Bile Acid-FXR Axis)

Dear Dr. Qiuyuo Hao:

Your manuscript has been accepted, and I am forwarding it to the ASM production staff for publication. Your paper will first be checked to make sure all elements meet the technical requirements. ASM staff will contact you if anything needs to be revised before copyediting and production can begin. Otherwise, you will be notified when your proofs are ready to be viewed.

Sincerely,
Neha Garg
Editor
mSystems

Reviewer #1 (Comments for the Author):

The authors replied to all questions. I have no further comments.